# Fabrication of Antibacterial Nanofibrous Membrane Infused with Essential Oil Extracted from Tea Tree for Packaging Applications

**DOI:** 10.3390/polym12010125

**Published:** 2020-01-05

**Authors:** Ji Yeon Lee, Joshua Lee, Sung Won Ko, Byeong Cheol Son, Jun Hee Lee, Cheol Sang Kim, Chan Hee Park

**Affiliations:** 1Department of Mechanical Design Engineering, Graduate School, Jeonbuk National University, Jeonju 54896, Korea; swc2736630@gmail.com (J.Y.L.); silvlise@hanmail.net (J.H.L.); 2Department of Bionanosystem Engineering, Jeonbuk National University, Jeonju 54896, Korea; jjangwon21@gmail.com (J.L.); vchamp21@naver.com (S.W.K.); hahaband@naver.com (B.C.S.); 3Division of Mechanical Design Engineering, Jeonbuk National University, Jeonju 54896, Korea

**Keywords:** nanofiber, tea tree oil, anti-bacterial, mechanical enhancement, CO_2_ reduction, packaging

## Abstract

Nanofibers made by electrospinning are being applied to an unlimited number of applications. In this paper, we propose the fabrication of antimicrobial functional nanofibers infused with essential oil for packaging applications that can extend the shelf-life of fruits. The morphology of nanofibers with different concentrations of essential oil was characterized by SEM and mechanical enhancement was confirmed via universal testing machine (UTM). The surface chemistry and crystalline of the nanofibers were investigated by FTIR and XRD, respectively. The CO_2_ reduction study was carried out using a hand-made experimental apparatus and nanofiber hydrophobicity, which can prevent moisture penetration from the outside, was evaluated by contact angle. Antimicrobial properties of the functional nanofibers were estimated by using Gram-negative/positive bacteria. The cytotoxicity of the functional nanofibers was studied using fibroblast cells. Furthermore, this study investigated how long the shelf-life of tomatoes was extended. The nanofibers could serve as a multifunctional packaging, as an emerging technology in agricultural products, and even contribute to a better quality of various distributed agricultural products.

## 1. Introduction

Electrospinning, which was first used in 1934 and has been actively studied for over 80 years, is a facile method used to produce nano-sized structures in which a high voltage is applied to an electrically charged polymer solution to form a Taylor cone by mutual electrostatic repulsion between surface charges and Coulombic force [1,2,3,4]. Nanofibers produced by electrospinning process have smaller diameters and higher specific surface areas per unit volume as compared with other technologies, so fibers produced by this method can be applied to a variety of fields [5,6,7,8]. Particularly in recent years, as the interest in nanometer-scale materials and their characteristics has increased, various studies using electrospinning have been rapidly increased [9].

Use of tea tree oil (TT oil) extracted from *Melaleuca alternifolia* [10] began in the 1770 s when the British explorer James Cook used it to prevent scurvy after seeing the Australian indigenous people use it to treat illnesses. TT oil has a high content of terpene, a volatile substance that destroys bacteria, and also has strong antifungal, antibacterial and antiviral properties [11,12]. It has been frequently used in dental and surgical treatment, and in particular, research and patents get raised with applications of TT oil and their constituents, and their fruit packaging incorporating, and are rapidly emerging as research on food and food microorganisms or oxidants [13,14,15,16]. However, using large quantities of undiluted TT oil can induce skin irritation and also reactions like allergies, rashes, dizziness and swelling of the throat, because the 1,8-cineole contained in the oil is toxic and as such, should be used in small amounts for local area applications [14,17,18]. Therefore, our work will contain a small amount of TT oil to prevent side effects and using the nanofibrous membrane infused with a small amount of TT oil, which can enlarged surface ratio for enhancing the antibacterial effect to prolong the fruits’ self-life.

In this study, we applied functional nanofibers to packaging to study the reduction of discharged carbon dioxide (CO_2_), the prevention of external moisture infiltration, as well as the prevention of decay by bacteria of the fruit through proper surrounding environment [19,20]. These nanofibers with essential oil also exhibit improved mechanical properties and are harmless to the human body. Therefore, we can expect these multifunctional nanofibers, which have antimicrobial properties, a CO_2_ reduction effect and moisture prevention effect, to be applied in a variety of other fields, such as packaging, filters and biomedical application.

## 2. Materials and Method

### 2.1. Preparation of Packaging Samples

Before proceeding with the experiment, the sample was prepared as shown in Figure 1A. More specifically, 12 wt% polyurethane (PU, Estane^®^, Skythane^®^ X595A-11, Mw = 110,000, State of Ohio, USA) solutions dissolved by dimethylformamide (DMF, 99.5%, Samchun chemical, Pyeongtaek, Korea) and tetrahydrofuran (THF, 99.8%, Samchun chemical, Pyeongtaek, Korea) in 1:1 volume ratio which containing 0, 1%, 3% and 5% of TT oil were prepared, to which a voltage of 15 kV was applied to produce nanofibers by electrospinning. To prepare the packaging container for study, after punching a circular hole of 5 cm in diameter in the packaging container, we attached the nanofibers to complete the sample of the packaging container.

### 2.2. Characterization

The morphology of the nanofibers according to the oil content was observed by scanning electron microscope (SEM JSM-5900, JEOL, Tokyo, Japan). The histogram of fiber diameter was drawn by measuring more than 50 nanofibers randomly from the SEM images using Image J program. The mechanical properties of the nanofiber were obtained using a universal test machine (UT-020E, MTDI INC, Daejeon, Korea) using a dog bone specimen according to the ASTM D638 protocol [21,22,23,24]. The functional groups and crystalline of surface chemistry were examined by Fourier transform infrared spectrometer (Frontier, Perkin Elmer Co., Waltham, The Commonwealth of Massachusetts, USA) and multi-purpose high performance X-ray diffractometer (X’pert Pro Powder, PANalyrical, Almelo, The Netherlands) to compare PU nanofibers with PU-5% TT oil sample. The wettability of the nanofibers was determined by contact angle meter (GBX, Digidrop, Bourg-de-Peage, France) at ambient temperature and measured at 1, 5, 10 and 90 s interval after 5 µL of water droplet was dropped onto the nanofiber’s surface. The carbon dioxide (CO_2_) reduction test was carried out through a hand-made experimental apparatus. This hardware of hand-made apparatus consists of a sensor part, chamber and CO_2_ circulator part. The software was based on the LabVIEW program.

### 2.3. Antibacterial Test

In vitro antibacterial activity of the samples was determined by growth inhibition study method using *Escherichia coli* (*E. coli*) and *Staphylococcus aureus* (*S. aureus*) [25]. Three types of bacterial colonies were thawed in lysogeny broth (LB) and were incubated separately for overnight in a shaking incubator at 37 °C. Using the spread plate method, after mixing 1 mL of thawed bacterial suspension containing around 10^5^ colony forming units for each bacteria with 9 mL of LB solution. The bacterial solution was suspended on the nutritive agar plate, and the samples were carefully placed on the inoculated plates. The agar plates with samples were incubated at 37 °C for 24 h and all antibacterial tests were performed in triplicate. The inhibition area measurement was executed by using ImageJ software. The zone of inhibition follows a standard.

### 2.4. In Vitro Biocompatibility Test

The cytotoxicity of the samples was estimated using fibroblast cells (NIH-3T3). The cells were cultivated in the cell culture plate with Dulbecco’s modified Eagles medium (DMEM, Sigma Chemical Co., St. Louis, MO, USA) including 10% fetal bovine serum (FBS, Sigma Chemical Co., St. Louis, MO, USA) and 1% penicillin/streptomycin in the incubator at 37 °C with 5% CO_2_ and 95% humidity. Prepared specimens were fixed in the wells of a 48-well cell culture plate (SPL life science, Korea) and sterilization under UV radiation for overnight. The cells with a density of 20,000 cells/well were seeded on the samples by pipetting onto the center. Each sample was cultured with cells for 1, 3 and 5 days and evaluated using cell counting kit-8 (CCK-8) in a 48-well plate.

### 2.5. Evaluation of Efficacy Using Tomatoes

To observe the degree of spoilage between the three different types of packaging, six tomatoes were prepared. The samples were randomly assigned and stored in each type of packages. The samples were stored for 14 days at around 30 °C.

### 2.6. Image and Statistical Analysis

All images were analyzed with ImageJ software. Results are presented as the mean ± standard deviation (SD) for *n* = 3. Statistical significance has assessed the level of significance with a one-way ANOVA followed by a Tukey test for means comparison by employing the OriginPro 8.5 software (*p* < 0.05).

## 3. Results and Discussion

Nanofibers can be applied to a variety of applications due to their many small pores and large surface area. In this study, we fabricated functional nanofibers and used it in packaging applications. In order to find an appropriate concentration of TT oil, we compared the morphology and mechanical strength of the nanofibers with different concentrations of TT oil. As can be seen from Figure 1B–I, as the content of TT oil gradually increased, the ratio of nano-sized fibers increased and the thickness of the micro-sized fibers became thinner. Therefore, the number of nanofibers with a diameter of one micrometer and the number of nanofibers with a diameter of 200 nanometers are about the same in the 5% TT oil group (Figure 1J–M). At the same time, the mechanical test results (Figure 1N) show that the nanofibers containing TT oil exhibit better strength and strain values than the PU nanofibers. As the TT oil content increased to 1%, 3% and 5%, the stress and strain values in breakpoint increased to 10.316, 11.931 and 12.319 MPa and 1329%, 1409.38% and 1455.07%, respectively. These results show that when TT oil was added, the respective values increased by about two times (PU stress and strain were 7.574 MPa and 793.83%, respectively). In addition, compared to the PU nanofiber, the nanofiber with TT oil tends to decrease the young’s modulus uniformly, which means the deformation of the nanofibers occurs relatively easily, allowing them to withstand more force. As can be observed from the SEM images, it can be expected that the increase in the number of nanofibers proportionally affected the mechanical strength. TT oil composition is given in the International Standard (ISO 4760-2004) and the main components of TT oil are terpinen-4-ol, γ-terpinene and α-terpinene and their contents are around 30%–48%, 10%–28% and 5%–13% respectively. So, pure TT oils have been reported to have peaks at 780, 799, 815, 830, 864 and 889 cm^−1^ [26]. Comparing Figure 2A, there are no differences between the chemical bonding of nanofibers with and without TT oils. This is because clear peaks are not shown due to the volatility of terpinene and terpinen-4-ol, and the peaks that appear by chance are hidden by the peaks of PU. The X-ray diffraction analysis was employed to identify the phase transformations and probable reaction due to the infusion of TT oil. As shown in Figure 2B, XRD patterns of the nanofibers with and without 5% TT oil reveals diffraction peak at the same 2θ, and this means there is no phase transformation and reaction because of infusing with TT oil.

The bacterial experiment was carried out using *Escherichia coli* (*E. coli*; Gram-negative) and *Staphylococcus aureus* (*S. aureus*; Gram-positive) to determine the antimicrobial properties of nanofibers containing 5% of TT oil [27,28]. In Figure 3A,B, the nanofibers containing TT oil (PU-TT oil) clearly showed a larger zone of inhibition compared to the control (PU). As shown in Figure 3C,E, it can also be seen that the inhibition zones of the TT oil groups were about three times greater than the control group. The results of CCK-8 in Figure 3D show that the cell proliferation of each sample in 1, 3 and 5 days were similar with the control (without nanofiber) group. This means that each of the nanofibers produced could be used as a packaging by showing it is not toxic to the human body.

To maintain the fruit in good condition, the air composition inside the packaging is also important. To measure the concentration of CO_2_ in packaging, one of the major factors of the proper internal air to extend the shelf-life of the fruit, we made the device depicted in Figure 4A–D and using it, we compared the CO_2_ reduction of the control (no nanofibers) with our prepared nanofibers (PU-5% TT oil). Figure 4E has three major facts. First, when the nanofibers were not installed, at 240 s when CO_2_ was introduced CO_2_ increased immediately, whereas when nanofibers were installed, the inflow of CO_2_ started with a smaller amount after a few seconds delay. Second, comparing the tilt graph of carbon dioxide inflow over time, it could be seen that the slope of the increase of CO_2_ when the nanofiber was installed was low. Finally, when the nanofibers were used, the increase of CO_2_ was stopped before the completion of the CO_2_ inflow. All of these facts mean PU-5% TT oil sample could be reduced the entrance of carbon dioxide into the package and it could maintain a proper environment for fruits. Therefore, the use of PU-5% TT oil was expected to reduce the amount of CO_2_ in the fruit packaging, and thus help the fruit maintain its initial state.

The contact angle results of the nanofibers are shown in Figure 4F. PU nanofibers with 5% TT oil (PU-TT oil), especially, has very similar results to the control. As the amount of TT oil in the nanofiber increased, it was observed that the hydrophobicity slightly decreased, but did not affect the overall hydrophobic nature of the nanofibers. Therefore, it is expected that the addition of tea tree oil will help prevent water from penetrating inside. Finally, the results of leaving the tomatoes in an environment of about 30 °C for a total of 14 days are shown in Figure 4G. On day 7, we were able to observe the decay of the tomatoes in the control group, especially in the top view, and on day 14, we observed the spoiled control group and slightly rotten PU group. Even after 14 days, the tomatoes with PU-TT oil samples were fresh, as confirmed by cutting open the tomatoes.

## 4. Conclusions

In summary, we prepared nanofibers based on tea tree essential oil by the electrospinning technique with antimicrobial, enhanced mechanical and moisture protection properties that could prolong fruit shelf-life. The characterizations and evaluations of all functional nanofibers were verified by experiments such as mechanical test, CO_2_ reduction test and antibacterial test. Especially, antibacterial tests proved that tea tree oil could still have antimicrobial properties even if it was fabricated into a nanofiber. In addition, the biocompatibility, one of the most important factors in packaging application, was tested by CCK-8 assay using fibroblast cells. In order to test whether the functional nanofiber was actually effective, an observational examination was carried out by storing them at room temperature for 14 days using tomatoes. Furthermore, nanofibers infused with TT oil were expected to be applied not only to fruit packaging, but also to other applications requiring antimicrobial activity such as filter and medical applications.

## Figures and Tables

**Figure 1 polymers-12-00125-f001:**
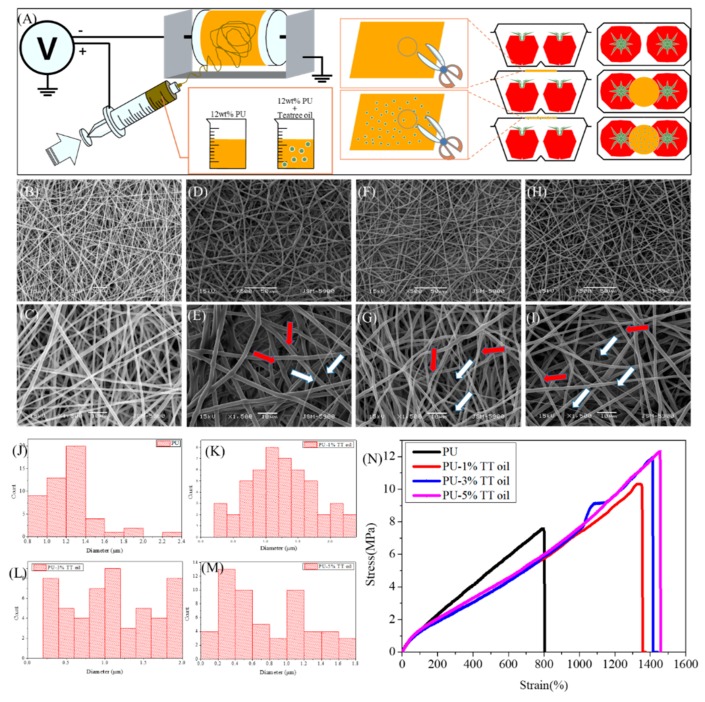
(**A**) The schematic of the packaging using electrospinning method. (**B**–**I**) SEM images in two magnitudes (×0.5 k (top), ×1.5 k (bottom)) of nanofibers with %1, 3% and 5% of tea tree (TT) oil (the red and white arrow indicates the nanofiber with diameter of 1 µm and 200 nm, respectively). (**J**–**M**) Histogram of diameter of nanofibers with 1%, 3% and 5% of TT oil. (**N**) Stress–strain curve of nanofibers with different concentrations of TT oil.

**Figure 2 polymers-12-00125-f002:**
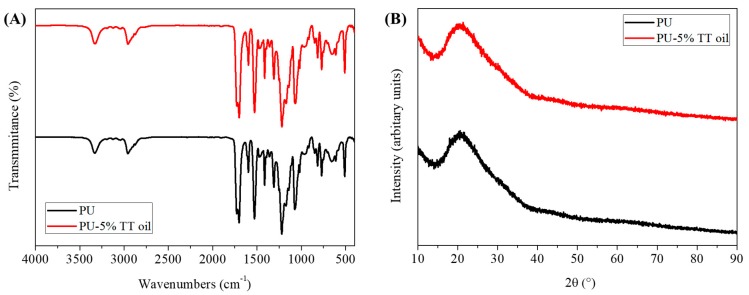
(**A**) Fourier transform infrared spectroscopy (FTIR) spectra and (**B**) X-ray diffraction (XRD) patterns of the PU and PU-5% TT oil.

**Figure 3 polymers-12-00125-f003:**
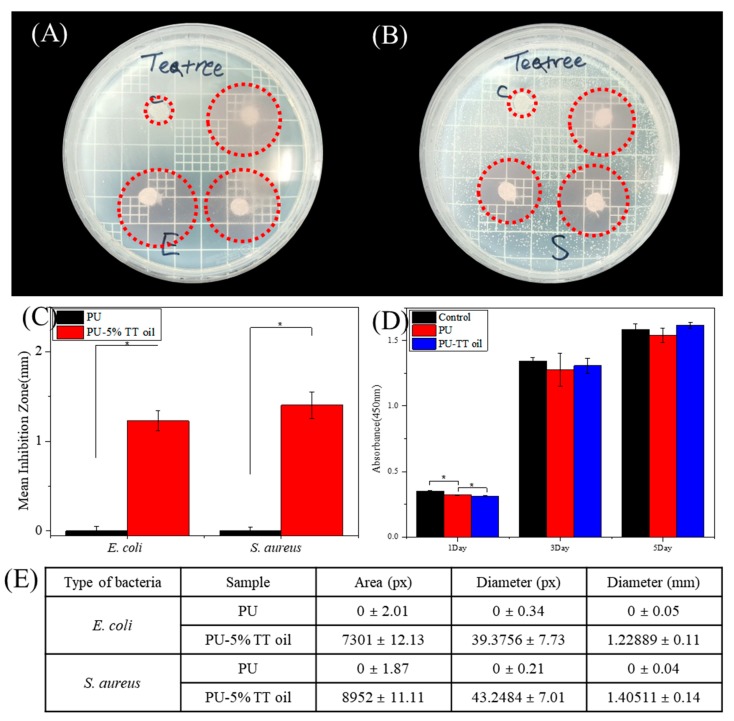
Representative images of the zone of inhibition using (**A**) *Escherichia coli* and (**B**) *Staphylococcus aureus* after 24 h after adhered samples. (**C**) The graph of inhibition zone diameter by Image J software. The data represent the mean three standard deviations with statistical significance. (**D**) Cytotoxicity test by CCK-8 assay on NIH-3T3 cell culture onto each sample at day 1, 3 and 5. (**E**) Comparison of antibacterial effect of PU and PU-5% TT oil using different type of bacteria (* indicates statistical significance (*p* < 0.05) measured by a one way ANOVA Tukey test).

**Figure 4 polymers-12-00125-f004:**
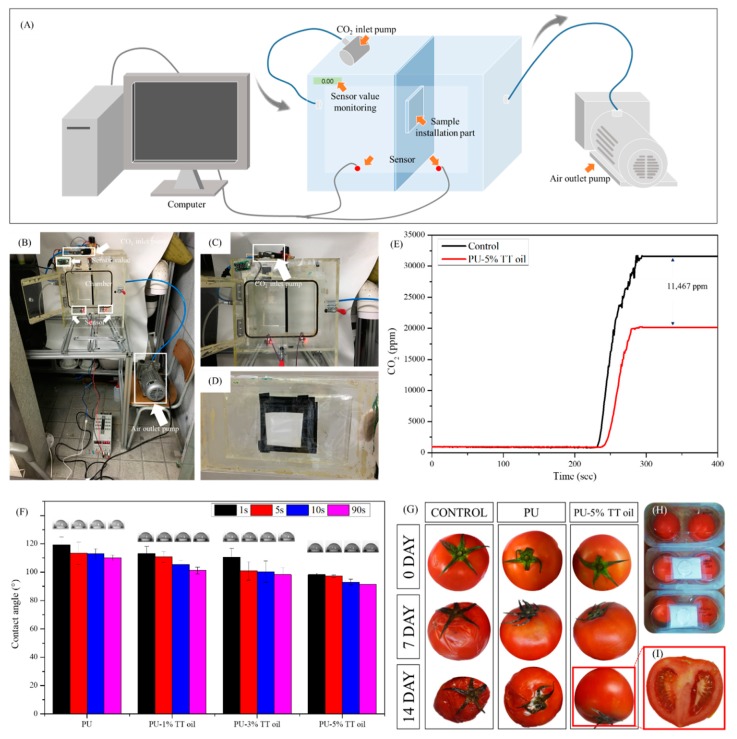
(**A**) Schematic of the CO_2_ concentration testing hand-made apparatus set-up. (**B**–**D**) Set-up of the hand-made apparatus for comparing the concentration of CO_2_. (**E**) The graph of the CO_2_ reduction study. (**F**) Contact angle values of nanofibers. (i–xii) Digital images of contact angle. (i, ii, iii) PU, (iv, v, vi) PU-1% TT oil, (vii, viii, ix) PU-3% TT oil and (x, xi, xii) PU-5% TT oil at 1, 5, 10 and 90 s. (**G**) Degree of spoilage between three different types of nanofiber for packaging using tomato. (**H**) Digital image of packaging. (**I**) The cross-section of tomato packed by PU-TT oil after 14 days of storage.

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
