# Peer review of "Fabrication of Antibacterial Nanofibrous Membrane Infused with Essential Oil Extracted from Tea Tree for Packaging Applications"

_polymers, 2020, doi:10.3390/polym12010125_

Round 1
Reviewer 1 Report
The authors have revised the manuscript and satisfactorily replied to all the points raised by the reviewer.
Author Response
The authors have revised the manuscript and satisfactorily replied to all the points raised by the reviewer.;
I'm really glad to hear that the reviewer gives me a good comment about the paper.

Reviewer 2 Report
Thank you for inviting me to evaluate the article titled “Fabrication of Antibacterial Nanofibrous Membrane Infused with Essential Oil Extracted from Tea Tree for Packaging Applications”, this paper describes an effective method of fabricating antibacterial nanofibrous membrane with excellent mechanical properties and good antibacterial activity. Thus I think this manuscript could be accepted after minor revision. The detail comments are listed following:
There is no explanation for increased strength and elongation at break Infrared image does not successfully characterize material composition, It is not a good idea put it in this paper. It is recommended to use a standard zone of inhibition for antibacterial testing.
Author Response
Thank you for inviting me to evaluate the article titled “Fabrication of Antibacterial Nanofibrous Membrane Infused with Essential Oil Extracted from Tea Tree for Packaging Applications”, this paper describes an effective method of fabricating antibacterial nanofibrous membrane with excellent mechanical properties and good antibacterial activity. Thus I think this manuscript could be accepted after minor revision. The detail comments are listed following;
I appreciate getting a chance to improve my paper and thank you for taking the time to evaluate the paper.
There is no explanation for increased strength and elongation at break.;
As commented by the reviewer, the explanation of strength and elongation at breakpoint was added in more detail to the results and discussion section.
Infrared image does not successfully characterize material composition, it is not a good idea put it in this paper.;
As the reviewer said, FTIR was not able to perfectly characterize the material composition. However, If adding FTIR and XRD, which are the basic analysis for characteristics, we can notify chemical characteristics such as chemical bonding and their properties. Moreover, the results of FTIR and XRD in the paper mean that even if TT oil is blended in nanofibers, the material's inherent constituent properties do not change. Therefore, I think these results make sense.
It is recommended to use a standard zone of inhibition for antibacterial testing.;
As you asked, we changed the graph to the standard zone of inhibition and added a table to identify this quantitatively in Fig 3.

This manuscript is a resubmission of an earlier submission. The following is a list of the peer review reports and author responses from that submission.
Round 1
Reviewer 1 Report
[1]. PU solutions in what solvent?
[2]. https://www.astm.org/Standards/D638.htm
"ASTM D638 - 14 Standard Test Method for Tensile Properties of Plastics
1.2 This test method is applicable for testing materials of any
thickness up to 14 mm (0.55 in.).
However, for testing specimens in the form of thin sheeting,
including film less than 1.0 mm (0.04 in.) in thickness,
ASTM standard D882 is the preferred test method.
Materials with a thickness greater than 14 mm (0.55 in.)
shall be reduced by machining."
You used "a dog bone specimen based on the ASTM D638 protocol."
but for thin specimens, the ASTM D638 protocol recommends to apply ASTM standard D882
Q: what means "based" ? Shouls I understand it is an adaptation ofthe D638 protocol?
Q: how thin were your specimens?
Q: how did you prepare you dog bone specimens from the microfiber mesh?
[3]. "The carbon dioxide (CO2) reduction test was carried out through a hand-made experimental
apparatus as shown in Fig 3. This hardware of hand-made apparatus is consist of a sensor part,
chamber, and CO2 circulator part."
BUT in Figure 3a I see "Air circulation pump"
Q: is it "CO2 circulator part" or "Air circulation pump" ?
[4]. "When PU-5% TT oil was used in passing the same amount of CO2, the amount of CO2 was
reduced to two-thirds compared to the case of control. Therefore, the use of PU-5% TT oil is
expected to reduce the amount of CO2 in the fruit packaging, and thus help the fruit maintain its
initial state. "
Q: how do you explain the CO2 reduction?
[5]. Figure 3D dificult to decipher:
-from 0-240 sec happens nothing , no CO2 in the chamber...if I understand correctly no CO2
pumped yet
-from 240-300 secs the CO2 concentration increases abruptly.....if I understand correctly CO2 is
pumped into the chamber????.
-from 300-400 secs the CO2 concentration remains constant
Q: as I can see, the process of CO2 reduction takes place in a 60 sec time interval (from 240-300
sec). How do you explain that?
Q: as I understand, the CO2 reduction was caused by "the use of PU-5% TT oil" . Please explain how
does the oil reduce de CO2 amount in such a short time interval (from 240-300 sec) ?
Q: you should add in the Figure 3D also the "PU" sample
Q: please explain the Figure 3D
Q: please explain the hand-made set-up for measuring the reduction of CO2 reduction.
Q: did you calibrate the sensor? how did you do it?
Q: what kind of sensor did you used?
Q: in Figure 3D how many replicates did you perform
Q" in Figure 3D you have the 11467 ppm value. What is the error of this value? how many replicates
did you used to obtain this value?
Q: in Figure 3D you have approx the 20000 ppm value for the PU-5% TT oil sample. What is the
error of this value? How many replicates did you used to obtain this value? The same question for
the control sample
[6]. "It was confirmed that microfibers with diameters of around 1 µm and nanofibers with
diameters of around 200 nm coincide especially in the 5% TT oil group."
Q: what do you mean by "conincide"?
Q: please indicate on the SEM figures where the 200 nm fiber are
Author Response
Reviewer 1
PU solutions in what solvent?.;
We were using Dimethylformamide (DMF) and Tetrahydrofuran (THF) in a 1:1 volume ratio as a solvent to dissolve the Polyurethane (PU). And this information was added in the revised manuscript in materials and method section.
https://www.astm.org/Standards/D638.htm
"ASTM D638 - 14 Standard Test Method for Tensile Properties of Plastics 1.2 This test method is applicable for testing materials of any thickness up to 14 mm (0.55 in.). However, for testing specimens in the form of thin sheeting, including film less than 1.0 mm (0.04 in.) in thickness, ASTM standard D882 is the preferred test method. Materials with a thickness greater than 14 mm (0.55 in.) shall be reduced by machining." You used "a dog bone specimen based on the ASTM D638 protocol." but for thin specimens, the ASTM D638 protocol recommends to apply ASTM standard D882.
Many papers used ASTM D638 to measure the mechanical properties of nanofiber specimens, so we decided that this method is suitable for our nanofiber specimens. And some of them are shown below as references.
Hsu, Y.H.; Lin, C.T.; Yu, Y.H.; Chou, Y.C.; Liu, S.J.; Chan, E.C. Dual delivery of active antibactericidal agents and bone morphogenetic protein at sustainable high concentrations using biodegradable sheath-core-structured drug-eluting nanofibers. Int J Nanomed 2016, 11, 3927-3937, doi:10.2147/Ijn.S107250. Kim, B.S.; Park, K.E.; Kim, M.H.; You, H.K.; Lee, J.; Park, W.H. Effect of nanofiber content on bone regeneration of silk fibroin/poly(epsilon-caprolactone) nano/microfibrous composite scaffolds. Int J Nanomed 2015, 10, 485-502, doi:10.2147/Ijn.S72730. Rogalski, J.J.; Zhang, H.; Yao, J.; Bastiaansen, C.W.M.; Peijs, T. High-modulus rotary jet spun co-polyimide nanofibers and their composites. Nanocomposites 2019, 10.1080/20550324.2019.1687174, doi:10.1080/20550324.2019.1687174. Seo, D.K.; Jeun, J.P.; Bin Kim, H.; Kang, P.H. Preparation and Characterization of the Carbon Nanofiber Mat Produced from Electrospun Pan/Lignin Precursors by Electron Beam Irradiation. Rev Adv Mater Sci 2011, 28, 31-34.
Also, we added these referenced papers as a reference [19-22] in materials and method section.
Q: what means "based"? Should I understand it is an adaptation of the D638 protocol?;
Sorry to confuse you for not using the accurate words. We used “based on” in the sense of “according to”, which means we experiment through the ASTM D638 protocol. This had also been corrected in the revised manuscript.
Q: how thin were your specimens? ;
For the mechanical property test, we were using the 3 specimens for each group. The thickness, width, gage length, and dimension of our specimens were shown in the table below.
|
Name of sample |
Thickness [mm] |
Width [mm] |
Gage length [mm] |
Dimension |
|
PU-1 |
0.035 |
3.18 |
9.53 |
0.1113 |
|
PU-2 |
0.035 |
3.18 |
9.53 |
0.1113 |
|
PU-3 |
0.030 |
3.18 |
9.53 |
0.0954 |
|
PU + 1% TT oil-1 |
0.052 |
3.18 |
9.53 |
0.16536 |
|
PU + 1% TT oil-2 |
0.020 |
3.18 |
9.53 |
0.0636 |
|
PU + 1% TT oil-3 |
0.030 |
3.18 |
9.53 |
0.0954 |
|
PU + 3% TT oil-1 |
0.032 |
3.18 |
9.53 |
0.10176 |
|
PU + 3% TT oil-2 |
0.019 |
3.18 |
9.53 |
0.06042 |
|
PU + 3% TT oil-3 |
0.042 |
3.18 |
9.53 |
0.13356 |
|
PU + 5% TT oil-1 |
0.043 |
3.18 |
9.53 |
0.13674 |
|
PU + 5% TT oil-2 |
0.009 |
3.18 |
9.53 |
0.02862 |
|
PU + 5% TT oil-3 |
0.035 |
3.18 |
9.53 |
0.1113 |
Q: how did you prepare you dog bone specimens from the microfiber mesh?.;
We have a custom UTM sampling cutter as follows, which is according to ASTM D638. The constant sized dog bone-shaped specimens can easily be prepared by placed in the instrument and pressed down.
(Left) Overall view of the UTM sampling device, (Right) cutter part.
"The carbon dioxide (CO2) reduction test was carried out through a hand-made experimental apparatus as shown in Fig 3. This hardware of hand-made apparatus is consist of a sensor part, chamber, and CO2 circulator part." BUT in Figure 3a I see "Air circulation pump"
Q: is it "CO2 circulator part" or "Air circulation pump"?;
Unfortunately, this is a problem caused by the incorrect translation. First, the part of the existing “air circulation pump” should be corrected to the “air outlet”, and the CO2 incoming part is connected to the left chamber as shown in the modified figure 4, and the CO2 is continuously introduced. This serves to suck air from the “outlet” connected to the right chamber to allow air to circulate from the left to the right chamber. So we named this whole part as the CO2 circulation part. All of this is detailed in the newly modified figure 4.
"When PU-5% TT oil was used in passing the same amount of CO2, the amount of CO2 was reduced to two-thirds compared to the case of control. Therefore, the use of PU-5% TT oil is expected to reduce the amount of CO2 in the fruit packaging, and thus help the fruit maintain its initial state. "
Q: how do you explain the CO2 reduction?.;
By measuring the amount of CO2 in the chamber on the right side when the same amount of CO2 flows into the chamber on the left side, the amount of CO2 in the chamber on the right side according to with or without nanofibers can be compared. The chamber on the left was always equipped with a sensor to confirm that the same amount of CO2 was introduced. On the other hand, the chamber on the right was equipped with a sensor to measure the amount of CO2 as it passed through the nanofiber. As shown in figure 4 D, the chamber on the right side, when nothing was installed at 240 sec when CO2 was introduced, immediately began to increase its concentration, whereas when nanofibers were installed, the concentration of CO2 was an increase in an amount of 2/3 after the delay few seconds. Also, in the case of install the nanofibers, the increase of CO2 was stopped at a time earlier than 300 sec when CO2 inflow was completed. When nothing is installed, it is confirmed that the amount of CO2 is continuously increased until 300 sec. Finally, the CO2 adsorption capacity of the nanofibers was proved by reducing the amount of CO2 finally measured by more than 1/3 when compared according to with and without nanofibers.
Figure 3D difficult to decipher:
-from 0-240 sec happens nothing, no CO2 in the chamber...if I understand correctly no CO2 pumped yet.
-from 240-300 secs the CO2 concentration increases abruptly.....if I understand correctly CO2 is pumped into the chamber????.
-from 300-400 secs the CO2 concentration remains constant.
Q: as I can see, the process of CO2 reduction takes place in a 60 sec time interval (from 240-300sec). How do you explain that?;
This experiment is designed to start the inflow of CO2 at 240 seconds and stop the inflow of CO2 at 300 seconds, and during the 60-second period and before/after in which the CO 2 is introduced, the comparison of the CO 2 concentration is made.
Q: as I understand, the CO2 reduction was caused by "the use of PU-5% TT oil" . Please explain how does the oil reduce de CO2 amount in such a short time interval (from 240-300 sec) ?;
We speculate that the surface of nanofibers with a large specific surface area helps to absorb carbon dioxide. So, to demonstrate this, we compared the carbon dioxide concentration before, after and during the introduction of carbon dioxide.
Q: you should add in the Figure 3D also the "PU" sample;
Unfortunately, an additional experiment using PU nanofibers is impossible because of the limitation of budget constraints. We believe that the PU-5% TT oil sample plays a similar role with PU nanofiber sample in the adsorption of CO2 cause of their similar surface area and physical/chemical properties. Therefore, it can be replaced PU nanofiber sample by experimenting using PU-5%TT oil sample.
Q: please explain the Figure 3D;
As mentioned earlier, Figure 4 D has three major facts. First, when the nanofibers are not installed, at 240 sec when CO2 is introduced CO2 increases immediately, whereas when nanofibers are installed, the inflow of CO2 starts with a smaller amount after a few seconds delay. Second, comparing the tilt graph of carbon dioxide inflow over time, it can be seen that the slope of the increase of CO2 when the nanofiber is installed is low. Finally, when the nanofibers were used, the increase of CO2 was stopped before the completion of the CO2 inflow. All of these facts mean PU-5% TT oil sample can be reduce the entrance of carbon dioxide into the package and it can be maintain a proper environment for fruits. These explanations have been added in the revised manuscript.
Q: please explain the hand-made set-up for measuring the reduction of CO2 reduction.;
The hand-made set-up consists of the following and can measure the CO2 concentration. In detail, there is a separator having a constantly exposed area in which nanofibers are inserted between the sealed chambers, and CO2 gas passes through the separator in a vertical direction. The CO2 gas flows into the left side of the chamber to pass through the separator, and on the right side of the chamber, the gas is inhaled by a vacuum pump to generate a gas flow. Each tube is controlled by a solenoid valve, a speed controller, and the like, and the CO 2 gas is sufficiently diffused inside each chamber.
Q: did you calibrate the sensor? how did you do it?;
The input signal was calculated based on the datasheet provided by the manufacturer and used according to the procedure of the manual provided. After that, all the sensors were put in one chamber, and four sensors showing the same tendency when changing the concentration of CO2 gas were selected and used (Inlet: 2, outlet: 2).
Q: what kind of sensor did you used?;
An electrochemical sensor was used, and the product was processed and calibrated in a module form by the manufacturer.
Q: in Figure 3D how many replicates did you perform;
We were repeated three times for each sample and the data were averaged and graphed.
Q" in Figure 3D you have the 11467 ppm value. What is the error of this value? how many replicates did you used to obtain this value?;
We averaged the values from three iterations for each sample and found a difference of 11467 ppm.
Q: in Figure 3D you have approx the 20000 ppm value for the PU-5% TT oil sample. What is the error of this value? How many replicates did you used to obtain this value? The same question for the control sample.;
As we answered the previous question, we ran a total of three experiments for each sample and platted the averaged values.
"It was confirmed that microfibers with diameters of around 1 µm and nanofibers with diameters of around 200 nm coincide especially in the 5% TT oil group."
Q: what do you mean by "conincide"?;
What I want to say in this sentence is the number of nanofibers with a one micrometer diameter and 200nm diameter are about the same. The manuscript has been modified to make this easier to understand. And this is shown in the added histogram of nanofiber diameter (Fig. 1.J-M).
Q: please indicate on the SEM figures where the 200 nm fiber are;
In figure 1 E, G, and I, representative nanofibers with diameters of 200nm and 1µm are indicated by arrows of different colors, respectively (The red arrow indicates the nanofiber of 1µm and the white arrow means the nanofiber of 200nm).
Reviewer 2 Report
Comments to the author: Major revisions
This work fabricated an electrospun nanofribrous membrane for packing application. It is interesting to use bio-based teatree oil to reinforce the antibacterial performances considering the extension of the shelf-life of fruit. However, even though all current data can support this point, still something important characterization and detailed discussion are missing, without which this paper is more like an experimental record than a research article. Except for this, the overall quality of the manuscript is good, and to my opinion it can be considered for publication in Polymers with major revisions. Below are some suggestions.
As shown in Fig. 1, it is an antibacterial nanofibrous membrane of film that was infused with the teatree oil rather than just bundles of “nanofibers”, so the title “Fabrication of Antibacterial Nanofibers Infused with Essential Oil Extracted from Tea Tree for Packaging Applications” is worthy of further consideration.
LACK of the characterizations of “teatree oil”, at least one of FTIR, Dynamic light scattering, XRD and NMR should be determined so the functional groups, average size, crystalline features or chemical structures of it will be clear. This is very important for the reader to understand why the teatree oil makes the electrospun membrane antibacterially different here.
In Fig. 1B-G, there is no significance differences observed between SEM images of Nanofibrous membrane with 1, 3 and 5% of TT oil. Please replace C, E, G with images of larger magnification. Please quantify “the increased ratio of nano-sized fibers” and “the thinner thickness of the micro-sized fibers” along with the gradually increased content of TT oil. Besides, the scale bar in all SEM images is too blurred to identify. In addition, though the stress and strain of the teatree oil -reinforced groups were increased, the young’s modulus of which was correspondingly decreased, would be good to discuss.
For Fig. 2.A-B, to label “PU-TT oil” Fig. 2.C-D is misleading, which should be “PU- 5% TT oil” as shown in Fig. 1H; if PU-5% TT oil presented much enhanced performance over PU, than what happened on PU- 1% TT oil and PU- 3% TT oil? In Fig. 3E, all samples presented unobvious changed within 10 s. Please refer to Cellulose, 2018, 25(2): 1309-1318 and prolong the testing duration to 90 seconds, which can offer more persuasive result.
The Results and Discussion section is LACK of discussion, not only for the morphology, mechanical testing, but also for the bacterial experiment, CO2 barrier and contact angle measurement. All we can see are “good results” for the PU-5% TT oil electrospun membrane, but why it was so good is unclear.
Author Response
This work fabricated an electrospun nanofribrous membrane for packing application. It is interesting to use bio-based teatree oil to reinforce the antibacterial performances considering the extension of the shelf-life of fruit. However, even though all current data can support this point, still something important characterization and detailed discussion are missing, without which this paper is more like an experimental record than a research article. Except for this, the overall quality of the manuscript is good, and to my opinion it can be considered for publication in Polymers with major revisions. Below are some suggestions.;
We would like to thank the Reviewer for taking time in reviewing our manuscript. The following comments had been helpful in improving the quality of our work.
As shown in Fig. 1, it is an antibacterial nanofibrous membrane of film that was infused with the teatree oil rather than just bundles of “nanofibers”, so the title “Fabrication of Antibacterial Nanofibers Infused with Essential Oil Extracted from Tea Tree for Packaging Applications” is worthy of further consideration.;
We agree with the reviewer’s comment on changing the title from nanofiber to nanofibrous membrane. This had been changed in the revised manuscript.
LACK of the characterizations of “teatree oil”, at least one of FTIR, Dynamic light scattering, XRD and NMR should be determined so the functional groups, average size, crystalline features or chemical structures of it will be clear. This is very important for the reader to understand why the teatree oil makes the electrospun membrane antibacterially different here.;
We have added FTIR and XRD data in the revised manuscript to demonstrate the surface chemistry and crystalline of TT oil. And the obtained results were evaluated, compared and discussed taking into account the effect caused by TT oil in PU nanofiber. However, we can’t do all the suggested measurements such as dynamic light scattering and NMR due to limitation of budget constraint.
In Fig. 1B-G, there is no significance differences observed between SEM images of Nanofibrous membrane with 1, 3 and 5% of TT oil. Please replace C, E, G with images of larger magnification. Please quantify “the increased ratio of nano-sized fibers” and “the thinner thickness of the micro-sized fibers” along with the gradually increased content of TT oil. Besides, the scale bar in all SEM images is too blurred to identify. In addition, though the stress and strain of the teatree oil -reinforced groups were increased, the young’s modulus of which was correspondingly decreased, would be good to discuss.;
We corrected Fig 1 and add the histogram of nanofiber's diameter. The size of the SEM images was enlarged by changing the configuration of Fig 1 and the scale bar is visible as SEM images were enlarged. The micro-sized and nano-sized nanofibers were indicated by red and white arrows in SEM images to help understand, respectively. We were measuring the nanofiber diameter and make it as a histogram to show the quantify "the increased ratio of nano-sized fibers" and "the tendency to decrease the diameter of micro-sized nanofibers" along with the gradually increased content of TT oil. Also, we have added more insight into the explanation of mechanical property relate to young's modulus in the revised manuscript.
For Fig. 2.A-B, to label “PU-TT oil” Fig. 2.C-D is misleading, which should be “PU- 5% TT oil” as shown in Fig. 1H; if PU-5% TT oil presented much enhanced performance over PU, than what happened on PU- 1% TT oil and PU- 3% TT oil? In Fig. 3E, all samples presented unobvious changed within 10 s. Please refer to Cellulose, 2018, 25(2): 1309-1318 and prolong the testing duration to 90 seconds, which can offer more persuasive result.;
The figures have been modified and revised accordingly. And we would like to emphasize the antimicrobial effect of the nanofiber combined with TT oil and since nanofibers showed the best results when the morphology and mechanical properties of nanofiber contained 5% thus these experiments were conducted around PU-5% TT oil sample. Sadly, we can't do the suggested extended experiment about contact angle from the reviewer as our lab and facilities are limited.
The Results and Discussion section is LACK of discussion, not only for the morphology, mechanical testing, but also for the bacterial experiment, CO2 barrier and contact angle measurement. All we can see are “good results” for the PU-5% TT oil electrospun membrane, but why it was so good is unclear.;
We have added more discussion in the revised version of the manuscript.

Reviewer 3 Report
The manuscript by Ji Yeon Lee and coworkers reports about the production of electrospun nanofibers containing tea tree oil and their employment in food packaging to extend the shelf life of fruit. The authors describe the fabrication and characterization of the nanofibers containing a different percentage of TT oil. The morphology and the mechanical characterization were shown by SEM and UTM, respectively. The antimicrobial properties were investigated by growth inhibition tests using Gram-positive and Gram-negative bacteria. The cytotoxicity was tested by cell proliferation tests on fibroblast cells. WCA was used to measure the hydrophobicity of the fibers; also, CO2 reduction in the presence of the nanofibers was measured. Finally, the shelf life of vegetables using food packaging containing the nanofibers was tested and proven up to 14 days.
The manuscript has some severe flaws. The authors should address the following points to improve the quality of their investigation:
1) The Introduction should be strengthened citing some recent reviews and papers focused on the use of essential oil extracted from aromatic plants (see for instance Ribeiro-Santos et al, Trends in Food Science and Technology, 61, (2017) 132-14); also, specific literature about tee tree oil (TT) should be added to the reference list (see for instance Carson et al., Clinical Microbiology Review, Jan 2006, p-50-62). I believe that some more information about the composition of the oil could be beneficial for the readers. TT has been widely recognized for its anti-microbial properties and for this reason utilized as ailments agent. However, TT can be also toxic, especially if ingested. Therefore its employment in food packaging should be considered with caution. Obviously, this aspect should be highlighted and discussed.
2) Related to the previous comment, do the authors have an evaluation of the amount of the cyclic compound delivered to the food? Specific migration limit of the toxic substances from packages to food should be strictly observed, in order not to constitute a healthcare risk. Does the cytotoxicity test performed by the author fully answer this question?
3) The authors fabricate nanofibers containing 1, 3 and 5% of TT oil. Some of the experiments were carried out using the three concentrations; other experiments (bacterial tests, evaluation of efficacy with tomatoes) were carried out only with the 5% content of TT in the fibers. What is the reason for this choice? If the 1% and 3% were not effective in the antimicrobial and/or were not effective in prolonging the shelf life of the vegetables, this should be more clearly stated. Otherwise, also the nanofibers with lower content of TT had to be tested.
4) In Fig.1 the SEM images of the nanofibers are reported. The control (0% TT) is missing, and it should be compared with the other samples. The dimension of the fibers is not reported, and it is hard to estimate the difference based only on the images. The sentences in which the SEM results are explained are difficult to interpret (line 89-91). What do the authors mean by ‘ratio’? How is the thickness estimated from the SEM images? And what does it mean that the diameters ‘coincide’? A table with diameter values and standard deviation in the 4 cases would be helpful.
5) The bacterial inhibition experiment is not well explained in the methods (no details are given). The corresponding figure in the results should also be improved. The quality of panel 2A is not very good; it is difficult to see the inhibition areas. Please, improve also the figure legend, giving all the details.
6) The cytotoxicity test is also not well explained in the methods. Were the cells cultured on the nanofiber samples? Details about the culture are totally missing (cell density, culture medium). Also images of the cells are not provided.
7) To test the spoilage between 3 different types of packaging, how many different tests were carried out? Was the reproducibility tested? This is not specified.
8) In the mechanical measurements, stress and strain values are given without errors and with no significant digits. Please amend this.
Author Response
The manuscript by Ji Yeon Lee and coworkers reports about the production of electrospun nanofibers containing tea tree oil and their employment in food packaging to extend the shelf life of fruit. The authors describe the fabrication and characterization of the nanofibers containing a different percentage of TT oil. The morphology and the mechanical characterization were shown by SEM and UTM, respectively. The antimicrobial properties were investigated by growth inhibition tests using Gram-positive and Gram-negative bacteria. The cytotoxicity was tested by cell proliferation tests on fibroblast cells. WCA was used to measure the hydrophobicity of the fibers; also, CO2 reduction in the presence of the nanofibers was measured. Finally, the shelf life of vegetables using food packaging containing the nanofibers was tested and proven up to 14 days. The manuscript has some severe flaws. The authors should address the following points to improve the quality of their investigation.;
We really appreciate taking time in comments of the reviewer and the following comments had been an opportunity to further enhancing the quality of our work.
The Introduction should be strengthened citing some recent reviews and papers focused on the use of essential oil extracted from aromatic plants (see for instance Ribeiro-Santos et al, Trends in Food Science and Technology, 61, (2017) 132-14); also, specific literature about tee tree oil (TT) should be added to the reference list (see for instance Carson et al., Clinical Microbiology Review, Jan 2006, p-50-62). I believe that some more information about the composition of the oil could be beneficial for the readers. TT has been widely recognized for its anti-microbial properties and for this reason utilized as ailments agent. However, TT can be also toxic, especially if ingested. Therefore its employment in food packaging should be considered with caution. Obviously, this aspect should be highlighted and discussed.;
It was intended for the manuscript to be a short communication, hence we only made a brief introduction. In the modified manuscript, we added information and references which focus on the use of tea tree oil as the reviewer recommended to the introduction section.
Related to the previous comment, do the authors have an evaluation of the amount of the cyclic compound delivered to the food? Specific migration limit of the toxic substances from packages to food should be strictly observed, in order not to constitute a healthcare risk. Does the cytotoxicity test performed by the author fully answer this question?.;
The nanofiber membrane fabricated in our paper is used as a material for a part of the packaging as described in materials and method sections and Fig 1, therefore, it is expected that there will be little direct contact with fruits. However, in order to verify the cytotoxicity of the unexpected invasion into the human body, the in-vitro experiment was conducted through fibroblast (NIH 3T3) cells, the most commonly used cell in the cytotoxicity test. Also, since we were well aware of the toxicity of TT oil, we conducted all experiments by limiting the content of TT oil to less than 5% in forming nanofibers.
The authors fabricate nanofibers containing 1, 3 and 5% of TT oil. Some of the experiments were carried out using the three concentrations; other experiments (bacterial tests, evaluation of efficacy with tomatoes) were carried out only with the 5% content of TT in the fibers. What is the reason for this choice? If the 1% and 3% were not effective in the antimicrobial and/or were not effective in prolonging the shelf life of the vegetables, this should be more clearly stated. Otherwise, also the nanofibers with lower content of TT had to be tested.;
First, an optimized sample called PU-5% TT oil was identified through morphology verification through SEM and characterization of chemical bonds, crystalline, and mechanical properties. This was to make the experiment run efficiently, and at the same time, 5% of TT oil contained in the nanofibers was expected to have the least toxicity and the highest antibacterial activity. Therefore, the experiment of nanofibers containing 1 and 3% of TT oil was considered unnecessary and the test was not conducted.
In Fig.1 the SEM images of the nanofibers are reported. The control (0% TT) is missing, and it should be compared with the other samples. The dimension of the fibers is not reported, and it is hard to estimate the difference based only on the images. The sentences in which the SEM results are explained are difficult to interpret (line 89-91). What do the authors mean by ‘ratio’? How is the thickness estimated from the SEM images? And what does it mean that the diameters ‘coincide’? A table with diameter values and standard deviation in the 4 cases would be helpful.;
SEM images of PU nanofiber and the histogram of nanofiber diameter has been added in the revised manuscript (Fig. 1.B,C,J-M). And the histograms show that the dimension of the fiber, and it helps to estimate the difference between specimens. And we have changed the sentences which explanation about SEM results for easy to interpret as the reviewer requested. The thickness of the nanofibers in the SEM image was measured by the Image J program, and histograms could be drawn based on this. The meaning of a sentence containing "coincide" was that the number of nanofibers with a one-micrometer diameter and 200nm diameter are about the same. The table recommended by the reviewer is good for intuitively judging the tendency of nanofiber diameter for each specimen, but the histogram is added because it is considered to be more helpful.
The bacterial inhibition experiment is not well explained in the methods (no details are given). The corresponding figure in the results should also be improved. The quality of panel 2A is not very good; it is difficult to see the inhibition areas. Please, improve also the figure legend, giving all the details.;
The section has been modified and revised accordingly. And in consideration of the difficulty of seeing clearly in the figure, the inhibition areas were marked with a red circle in Fig. 3.A,B.
The cytotoxicity test is also not well explained in the methods. Were the cells cultured on the nanofiber samples? Details about the culture are totally missing (cell density, culture medium). Also images of the cells are not provided.;
This had been added in the revised version of the manuscript. Unfortunately, additional experiment using confocal laser scanning microscope (CLSM) to confirm cell morphology is impossible as limitation of budget constraint. But we thought that the graph of cell viability through cck-8 staining method alone can fully explain cytotoxicity.
To test the spoilage between 3 different types of packaging, how many different tests were carried out? Was the reproducibility tested? This is not specified.;
This is mentioned in the “2.5. Evaluation of Efficacy using Tomatoes section”.
In the mechanical measurements, stress and strain values are given without errors and with no significant digits. Please amend this.;
This graph is the average of the results of three times of each sample. There is no error because only the average value is displayed. The reason for graphing with average values alone is that if error values (standard deviation) were came out, otherwise the graphs will be more complex, making it difficult to evaluate physical property performance.

Round 2
Reviewer 1 Report
CITATION FROM LITERATUrE: Modified atmospheres (MA), i.e., ELEVATED concentrations of carbon dioxide and reduced levels of oxygen and ethylene, can be useful supplements to provide optimum temperature and relative humidity in maintaining the quality of fresh fruits and vegetables after harvest [Crit Rev Food Sci Nutr. 1989;28(1):1-30. Modified atmosphere packaging of fruits and vegetables. Kader AA1, Zagory D, Kerbel EL.].
BUT THE AUTHORS CLAIM : "All of these facts mean PU-5% TT oil sample can be reduce the entrance of carbon dioxide into the package and it can be maintain a proper environment for fruits."
AND "Therefore, the use of PU-5% TT oil is expected to reduce the amount of CO2 in the fruit 162 packaging, and thus help the fruit maintain its initial state."
================================
the home made device still not explained !!!!:
"This hardware of hand-made apparatus is consist of a sensor part, chamber, and CO2 circulator part."-THIS IS NOT CLEAR AT ALL.
Where is the electrospun mesh? Is it in the chamber, in the wall of the chamber?
Reviewer 2 Report
The authors haven't replied all comments from me, and also some important tests were not conducted, such as NMR and dynamic water contact angle. So I don't think it is publishable.
Reviewer 3 Report
The revised version of the manuscript by Ji Yeon Lee and co-workers has been thoroughly improved and all the criticism of the reviewers have been addressed. The introduction has been strengthened, as required. The results have been completed with new data, including some controls. The data presentation is clearer. No further changes are requested.